# RAC1b Collaborates with TAp73α-SMAD4 Signaling to Induce Biglycan Expression and Inhibit Basal and TGF-β-Driven Cell Motility in Human Pancreatic Cancer

**DOI:** 10.3390/biomedicines12010199

**Published:** 2024-01-16

**Authors:** Hendrik Ungefroren, Julissa Reimann, Björn Konukiewitz, Rüdiger Braun, Ulrich F. Wellner, Hendrik Lehnert, Jens-Uwe Marquardt

**Affiliations:** 1First Department of Medicine, University Hospital Schleswig-Holstein (UKSH), Campus Lübeck, 23538 Lübeck, Germany; 2Institute of Pathology, University Hospital Schleswig-Holstein (UKSH), Campus Kiel, 24105 Kiel, Germany; 3Department of Surgery, University Hospital Schleswig-Holstein (UKSH), Campus Lübeck, 23538 Lübeck, Germany; 4University of Salzburg, 5020 Salzburg, Austria

**Keywords:** biglycan, cell migration, epithelial-mesenchymal transition, PDAC, RAC1b, SMAD3, SMAD4, TAp73, transforming growth factor-β

## Abstract

Pancreatic ductal adenocarcinoma (PDAC) is a highly aggressive cancer type characterized by a marked desmoplastic tumor stroma that is formed under the influence of transforming growth factor (TGF)-β. Data from mouse models of pancreatic cancer have revealed that transcriptionally active p73 (TAp73) impacts the TGF-β pathway through activation of Smad4 and secretion of biglycan (Bgn). However, whether this pathway also functions in human PDAC cells has not yet been studied. Here, we show that RNA interference-mediated silencing of TAp73 in PANC-1 cells strongly reduced the stimulatory effect of TGF-β1 on *BGN*. TAp73-mediated regulation of *BGN*, and inhibition of TGF-β signaling through a (Smad-independent) ERK pathway, are reminiscent of what we previously observed for the small GTPase, RAC1b, prompting us to hypothesize that in human PDAC cells TAp73 and RAC1b are part of the same tumor-suppressive pathway. Like TAp73, RAC1b induced SMAD4 protein and mRNA expression. Moreover, siRNA-mediated knockdown of RAC1b reduced TAp73 mRNA levels, while ectopic expression of RAC1b increased them. Inhibition of BGN synthesis or depletion of secreted BGN from the culture medium reproduced the promigratory effect of RAC1b or TAp73 silencing and was associated with increased basal and TGF-β1-dependent ERK activation. BGN also phenocopied the effects of RAC1b or TAp73 on the expression of downstream effectors, like the EMT markers E-cadherin, Vimentin and SNAIL, as well as on negative regulation of the ALK2-SMAD1/5 arm of TGF-β signaling. Collectively, we showed that tumor-suppressive TAp73-Smad4-Bgn signaling also operates in human cells and that RAC1b likely acts as an upstream activator of this pathway.

## 1. Introduction

Pancreatic ductal adenocarcinoma (PDAC) represents the most abundant type of pancreatic cancer and is characterized by early metastatic spread, late diagnosis and the lack of efficient therapies [1,2,3]. PDAC is predicted to become the second leading cause of cancer-related deaths worldwide by the year 2030. The extremely poor prognosis emphasizes the urgent need for obtaining a better understanding of PDAC development and progression. Recent studies have focused on the tumor stroma, which comprises the majority of the tumor mass [4,5] and is extremely rich in various extracellular matrix (ECM) components, a phenomenon termed desmoplasia [5]. Increasing efforts have been made to therapeutically target this non-malignant but still transformed compartment in order to slow down tumor development and reduce its aggressive nature [4,6]. Tumor stroma formation and composition are controlled to a large extent by transforming growth factor (TGF)-β. A prototype example for a matrix protein, whose expression is induced by this growth factor is the small leucine-rich proteoglycan, biglycan (BGN). Of note, the secreted form of BGN is able to bind and sequester TGF-β in the pericellular space, thereby preventing access to its cognate receptors and neutralizing its biological activity towards tumor-promoting effects. BGN and TGF-β form an autoregulatory feedback loop since *BGN* itself is subject to positive regulation by canonical TGF-β signaling involving the common-mediator Smad, SMAD4 [7]. The canonical TGF-β/Smad pathway involves, besides SMAD4, the receptor-regulated Smads, SMAD2 and SMAD 3 [8]. Alterations in this pathway, particularly mutations in or genomic deletion of *DPC4* (encoding SMAD4) are crucial steps in PDAC progression [4]. However, TGF-β can also signal through Smad-independent pathways, i.e., the extracellular signal-regulated kinases ERK1/2, JNK/p38 or PI3K/AKT [9,10,11]. The differential activation of canonical and non-canonical signaling by TGF-β, which in turn depends on the cellular context and disease stage, determines to a large extent whether this growth factor acts as a tumor suppressor or tumor promoter in PDAC [12].

Recently, it was demonstrated in two genetically engineered mouse models (GEMMs) of pancreatic cancer that the TGF-β pathway is controlled by full-length, transcriptionally active p73 (TAp73), a p53 family member and close homologue [13], through regulation of Bgn secretion via intermittent Smad3/4 expression/activity [14]. Removal of TAp73, and, as a consequence, deficient Smad3 and Smad4 expression, led to activation of TGF-β signaling through a Smad-independent pathway, favoring oncogenic TGF-β effects like epithelial-mesenchymal transition (EMT), migratory and invasive abilities and reduced survival [14]. The pro-EMT effect could be attributed to the loss of Bgn secretion and its function as an inhibitor of TGF-β, a master regulator of the EMT process. A simultaneous absence of TAp73 and Bgn led to a reinforced TGF-β signaling switch from Smad-dependent to Smad-independent pathways with activation of Erk and Pi3k signaling after treatment with TGF-β1. Extending these investigations to human PDAC in vitro has revealed that the α isoform of TAp73 has a similar role in human PDAC, blocking basal and TGF-β1-dependent activation of ERK1/2 and cell motility [15]. Mechanistically, this is a consequence of TAp73-induced induction of *DPC4* and subsequent SMAD4-mediated inhibition of ERK activation and cell migration [15]. However, it remained open if these SMAD4 effects are direct, or if *BGN* as a SMAD4 response gene is mediating them.

These newly discovered functions of TAp73 revealed in murine and human PDAC cells, namely induction of expression of Smads, Bgn/BGN and concurrent inhibition of Erk/ERK activation and cell motility are reminiscent of what we observed earlier for the small GTPase, RAC1b, a splice isoform of the *RAC1* gene, in PDAC cells of human origin [16,17,18,19]. Indeed, RAC1b promoted SMAD3 and BGN expression [16], and inhibited TGF-β1-induced ERK1/2 activation [17] and cell migration [17,18]. Moreover, our previous immunohistochemical and immunoblot data from human PDAC tissues and cell lines have shown that RAC1b is preferentially expressed in G1 and G2 but less in G3 tumors [18]. RAC1b is also more abundant in well-differentiated PDAC cells with an epithelial phenotype [16]. RAC1b shares many cellular responses associated with inhibition of EMT in common with TAp73 although in different cellular systems: promotion of the expression of epithelial markers such as E-cadherin and BGN, as well as inhibition of the mesenchymal markers SNAIL, vimentin and N-cadherin [16,17]. In addition, both RAC1b [16,17,18] and TAp73 [14,15] inhibited the tumor cells’ basal and TGF-β1-dependent migratory activity. In regard to RAC1b, we earlier identified SMAD3 as a mediator of this inhibitory effect [16] (Figure 1).

Thakur and colleagues [14] have almost exclusively focused on the murine system and hence it is not clear whether the above-mentioned effects of TAp73 in murine cells also operate in their human counterparts and whether they are duplicated by RAC1b, i.e., promotion of SMAD4 expression. When comparing these types of regulatory interactions, which we have illustrated in Figure 1, a conceivable scenario arose from it, namely that TAp73 and RAC1b may be part of the same tumor-suppressive pathway in human PDAC cells to sustain SMAD4 and BGN expression, while suppressing ERK1/2 activation and cell migration/invasion. Here, we studied if TAp73 regulation of BGN previously described in GEMMs also operates in human PDAC cells, and if RAC1b collaborates with TAp73 in this activity. However, in order to demonstrate that TAp73 and RAC1b synergize in inducing *BGN* and in inhibiting cell motility through BGN in human PDAC cells, several open questions need to be resolved in the human system: (i) does TAp73 induce BGN, (ii) does RAC1b induce SMAD4, (iii) does RAC1b activate TAp73 or vice versa and (iv) can BGN mimic the inhibitory effect of TAp73 or RAC1b on ERK activation and cell motility. Using various human pancreatic cancer cell lines we demonstrate here that tumor-suppressive TAp73-Smad4-Bgn signaling also operates in human PDAC and that RAC1b likely acts as an upstream activator of this pathway.

## 2. Materials and Methods

### 2.1. Reagents

In this study, we employed the following antibodies: anti-phospho-ERK1/2, #4370, Cell Signaling Technology (CST, Frankfurt am Main, Germany); anti-Smad4 (B8), #sc-7966; anti-HSP90 (F-8), #sc-13119, Santa Cruz Biotechnology (Heidelberg, Germany); anti-RAC1b, #09-271 (Merck Millipore, Darmstadt, Germany); anti-RAC1, #610650 (BD Biosciences, Heidelberg, Germany); anti-E-cadherin, #3195 (CST); anti-Snail, #3895 (CST); anti-Vimentin, #ab3974 (Abcam, Cambridge, UK,); anti-phospho-Smad1 (Ser463/465)/Smad5 (Ser463/465)/Smad8 (Ser426/428), #9511 and anti-phospho-Smad3 (Ser423/425), #9514 (both from CST). The secondary antibodies, HRP-linked anti-rabbit, #7074, and anti-mouse, #7076, were purchased from CST and recombinant human TGF-β1, #300-023, from ReliaTech (Wolfenbüttel, Germany). An antibody raised against a peptide within the mature form of human BGN (LF-51) was a kind gift from Dr. L. W. Fisher (NIDCR, National Institutes of Health). Prevalidated siRNA to p73 was purchased from Santa Cruz Biotechnology and others to RAC1b, SMAD3 or SMAD4 from Invitrogen/Thermo Fisher Scientific (Darmstadt, Germany). The HA-TAp73α and HA-TAp73β vectors were donated by Drs. B. Joseph and P. Engskog Vlachos (Stockholm, Sweden) and an expression vector for human BGN by L. Schäfer (Frankfurt am Main, Germany). The expression vector for human *TGFB1* was provided by OriGene Technologies Inc. (Rockville, MD, USA, #SC119746).

### 2.2. Cells and Transfection with SiRNA or Plasmid DNA

PANC-1 human PDAC cells were purchased from the ATCC (Manassas, VA, USA), while HPAFII and L3.6pl cells were supplied by Dr. U.F. Wellner. Cells were maintained in RPMI 1640 containing 10% fetal bovine serum (FBS), 1% Penicillin-Streptomycin-Glutamine (Thermo Fisher Scientific) and 1% sodium pyruvate (Merck Millipore). The generation and characterization of PANC-1 cells stably expressing either HA-RAC1b, or empty pCGN vector, have been described in detail earlier [18].

For transfection of siRNA or plasmid DNA, PANC-1, HPAFII or L3.6pl cells were seeded on day 1 and transfected the next day serum-free with 50 nM of prevalidated siRNAs specific for p73 [15], RAC1b [17], BGN [16] or scrambled siRNAs as control, with either Lipofectamine 2000 (PANC-1) or Lipofectamine RNAiMAX (HPAFII and L3.6pl) (both from Thermo Fisher Scientific) according to the manufacturer’s instructions. The pcDNA3.1-based expression vector for human BGN was introduced into PANC-1 cells using Lipofectamine 2000 according to the manufacturer’s protocol. Transfected cells were subjected to either qPCR analysis, immunoblotting or cell migration assays.

### 2.3. RT-PCR Analysis

Total RNA was extracted by affinity chromatography on columns (innuPREP RNA Mini Kit 2.0, IST Innuscreen GmbH, Berlin, Germany). The general RT-PCR protocol was described in detail earlier [17]. Briefly, 2.5 μg RNA was reverse transcribed for 1 h at 37 °C in a total volume of 20 μL with M-MLV Reverse Transcriptase (200 U) and random hexamers (2.5 μM) (both from Life Technologies/Thermo Fisher Scientific). Relative quantification of target genes by quantitative RT-PCR (qPCR) was done with Maxima SYBR Green Mastermix (Thermo Fisher Scientific) on an I-Cycler with Quant Studio Design & Analysis software, version 1.3.1 (BioRad, Munich, Germany). For data normalization, we also determined the expression of glycerinaldehyde-3-phosphate-dehydrogenase (*GAPDH*) and/or TATA box-binding protein (*TBP*). The sequences of PCR primers were given in previous publications [15,16,17,18], except for Laminin γ2 (LAM2C, forward: 5′-GGAAAGGAAGGAGCTGGAGT-3′, reverse: 5′-TGTTGATCTGGGTCTTGGCT-3′).

### 2.4. Immunoblotting

The immunoblotting procedure was described in detail in earlier publications [15,16,17,18]. An amount of 20–40 μg of total cellular protein quantified with the DC Protein-Assay Kit (BioRad) was fractionated by SDS-PAGE on mini-PROTEAN TGX any-kD precast gels (BioRad) and transferred to polyvinylidene difluoride (PVDF) membranes (Immobilon-P, Millipore, Eschborn, Germany) equilibrated with methanol and transferred to blotting buffer. After blotting, membranes were blocked with Tris-buffered saline containing 0.1% Tween 20 (TBST) and 5% bovine serum albumin. Following overnight incubation with the primary antibody at 4 °C in TBST, the primary antibody was removed by washing the blots with TBST. Subsequently, blots were incubated with the appropriate peroxidase-conjugated secondary antibodies and developed with the chemiluminescent detection kit (Amersham ECL Prime Detection Reagent, Cytiva, Marlborough, MA, USA) following the manufacturer’s protocol on a ChemiDoc XRS imaging system (BioRad). Signal quantifications for the proteins of interest and HSP90 were done by densitometry using either the built-in function of the ChemiDoc XRS system or the program Image Lab 5.2.1. The antibodies used are listed in Section 2.1.

### 2.5. Real-Time Cell Migration Assays

For the measurement of random/spontaneous cell migration in a chemokinesis setup, we used the xCELLigence^®^ DP system from ACEA Biosciences (San Diego, CA, USA) as outlined in detail earlier [16,17,18]. Following equilibration of the wells of the CIM plate-16 (OLS, Bremen, Germany) with standard growth medium at 37 °C for 1 h, a total of 60,000–80,000 cells (transfected before with various siRNAs or expression vectors as indicated in the figure legends) were loaded in the upper chamber of each well. To minimize proliferation, a standard growth medium supplemented with only 1% rather than 10% FBS was added along with the cells. Cells were then allowed to settle for 30 min prior to the start of the assay in an xCELLigence Real-Time Cell Analyzer DP device (Agilent Technologies, Santa Clara, CA, USA). Migration data were recorded every 15 min for different times and analyzed with RTCA software, version 1.2 (ACEA Biosciences). In some assays, anti-BGN antibody or isotype control antibody was added to the medium of the upper chamber prior to the start of the assay and remained there until assay termination.

### 2.6. Statistical Analysis

The statistical significance was assessed with the Wilcoxon rank-sum test using SPSS, version 26.0. *p*-values of <0.05 (*) were deemed significant.

## 3. Results

### 3.1. TAp73 Upregulates the Small Proteoglycan BGN and Is Required for Its Induction by TGF-β1 in Human PDAC Cells

Employing primarily PANC-1 and HPAFII cells we demonstrated previously that both RAC1b and TAp73 inhibited EMT through induction of *CDH1* (and other epithelial genes) and by suppression of mesenchymal genes and cell migration [15,17]. Moreover, production of the secreted proteoglycan Bgn, an inhibitor of TGF-β biological activity and thus an “anti-mesenchymal” gene, was shown in murine PDAC cells to be promoted by TAp73 and in the human orthologues by RAC1b, but whether this also applies to TAp73 in human PDAC cells is not known. To this end, siRNA-mediated knockdown of p73 in PANC-1 or HPAFII cells resulted in a dramatic decline in BGN mRNA (Figure 2A).

The p73 siRNA is likely to inhibit not only TAp73α but also TAp73β which in immunoblots possesses a higher mobility due to lack of the SAM domain [15,19]. To explore whether both TAp73 isoforms differ in their ability to induce *BGN*, we ectopically expressed both isoforms in PANC-1 or HPAFII cells using appropriate vectors. As shown previously, both isoforms are expressed and show the expected size difference [15]. When we monitored transfectants of both cell lines by qPCR analysis for BGN, we observed elevated mRNA levels only in TAp73α but not TAp73β transfected cells (Figure 2B). As a control, we have used primers for *DPC4*/SMAD4 [15], confirming positive regulation of both genes by TAp73α but not TAp73β.

We [15] and others [14] observed that the ability of TGF-β1 to induce luciferase activity through SMAD binding in PDAC cells was lost in TAp73-deficient PDAC cells, suggesting that TAp73 promoted TGF-β signaling by activating Smad proteins. If so, this should also impact the response of *BGN* to TGF-β1 stimulation in human cells, since we have previously shown that this gene is dramatically induced by this growth factor in a SMAD4-dependent manner [7]. To this end, in p73-silenced PANC-1 cells basal expression as well as the inductive effect of TGF-β1 on *BGN* after a 24 h stimulation period was reduced from 48.3 ± 6.7-fold to 8.8 ± 3.2-fold (Figure 2C). We conclude from these data that in human PDAC cells TAp73α is necessary for both basal expression of *BGN* and its full-blown response to TGF-β1.

### 3.2. RAC1b Induces Expression of SMAD4 at the mRNA and Protein Level

As part of a previous study, we showed that RAC1b—via SMAD3—promoted the expression of *BGN* [16], a gene previously shown by us to be regulated also by SMAD4 [7]. More recently, Thakur and coworkers have shown in GEMM-derived PDAC tissues and cell lines that Bgn expression is also upregulated by TAp73 through intermittent induction of Smad4 [14]. Given the SMAD4-dependency of the TGF-β effect on *BGN* [7], we therefore analyzed whether RAC1b, too, in addition to SMAD3, utilizes SMAD4 to induce *BGN*. For this purpose, we knocked down RAC1b in PANC-1 cells by transfection with a RAC1b-specific siRNA (previously validated in [18]), or a scrambled siRNA as control. In cells that received the RAC1b siRNA (termed PANC-1-RAC1b-KD), we observed a decrease in the abundance of SMAD4 protein (Figure 3A and Appendix A). We further reasoned that if RAC1b were to stimulate SMAD4 protein synthesis through TAp73 via binding to the p53 binding sites present in the *DPC4* promoter and a subsequent increase in *DPC4* transcriptional activity [14], an increase in SMAD4 mRNA levels would be expected. To test this, we quantified SMAD4 mRNA in PANC-1-RAC1b-KD (Figure 3B) and in L3.6pl-RAC1b-KD cells (Figure 3C), and in both lines observed a decrease in their abundance relative to controls. This suggests that RAC1b induces SMAD4 protein and mRNA expression, likely via TAp73 and TAp73 binding to the *DPC4* promoter.

### 3.3. Knockdown of RAC1b Downregulated TAp73 Expression but Not Vice Versa

In the previous section, we have shown that RAC1b shares in common with TAp73 the ability to induce the expression of *DPC4*. Moreover, RAC1b and TAp73 have an overlapping target gene spectrum in PDAC cells characterized by upregulation of epithelial genes and *BGN*, and downregulation of mesenchymal genes and pathways, i.e., MEK-ERK [15,17]. We, therefore, addressed the question of whether both proteins are part of the same pathway, and more specifically whether RAC1b is located up or downstream of TAp73. To look at this more closely, we carried out reciprocal inhibition and overexpression experiments. First, to reveal if TAp73 regulates RAC1b, we knocked down TAp73 in PANC-1 cells by RNAi followed by qPCR for exon 3b of *RAC1*, or by immunoblotting using a RAC1b-specific antibody. Knockdown of TAp73 was able to silence TAp73 but was unable to reduce RAC1b (or RAC1) protein levels (Figure 4A and Appendix A), or RAC1b mRNA abundance (Figure 4B). In contrast, the knockdown of RAC1b using an siRNA directed against exon 3b of *RAC1* [18] was able to reduce TAp73 mRNA levels by 58% relative to controls (Figure 4B).

To confirm these regulatory interactions, we employed PANC-1 cells with transient ectopic expression of TAp73α (see Figure 2B). Although in these cells as well as in cells transfected with TAp73β, RAC1b mRNA levels were marginally elevated over controls, this small increase was unlikely to be of physiological significance (Figure 4C). Conversely, ectopic expression of a HA-tagged version of RAC1b (HA-R1b) increased the mRNA levels of TAp73 by 1.55-fold over those in empty vector-transfected control cells (Figure 4D). When all findings were taken together, we concluded that RAC1b is located upstream of TAp73 in the TAp73-SMAD4-BGN pathway.

### 3.4. The Increase in Cell Migration upon RAC1b Knockdown Is Partially Rescued by Ectopic Expression of TAp73, While the Decrease in Cell Migration upon Ectopic Expression of RAC1b Is Partially Rescued by p73 Knockdown

In previous studies, we have already demonstrated that RAC1b [16,17,18], like TAp73α [15], strongly suppressed migratory activities in PANC-1 and other PDAC cells, lending further support to the notion that both proteins also collaborate in the negative control of cell motility. To reveal whether the upstream location of RAC1b relative to TAp73, postulated on the basis of the expression data in Section 3.3, also extended to a functional level, we performed mutual rescue experiments. To this end, PANC-1 cells were transfected with either a RAC1b siRNA in combination with HA-TAp73α expression vector or empty pcDNA3.1 vector, or, alternatively, HA-R1b-pCGN vector in combination with p73 siRNA. When we subjected these transfectants to real-time cell migration assays, we observed that ectopic HA-TAp73 but not empty vector partially prevented the RAC1b knockdown-induced rise in migratory activity (Figure 5A), while, conversely, the HA-R1b-pCGN-induced inhibition of migratory activity was partially prevented by RNAi-mediated p73 knockdown under both basal conditions and in the presence of added TGF-β1 (Figure 5B).

We then addressed the question of whether the combined knockdown of p73 and RAC1b would further enhance migratory activity in an additive or synergistic manner. However, as shown in Figure 5C, silencing TAp73 and RAC1b simultaneously did not provide an extra increase in migratory activity beyond that achieved with the RAC1b knockdown alone. However, sip73 alone was unable to provide the same rescue effect in the absence of siRAC1b (Figure 5C).

The increase in cell migration upon RAC1b knockdown was partially rescued by ectopic expression of TAp73. Since migration is related to EMT, we evaluated the levels of EMT markers (ECAD and SNAIL2/SLUG) on PANC-1 cells transfected with siRNA to RAC1b along with expression vectors for either TAp73α or TAp73β (corresponding to panel A). It turned out that TAp73α, but not TAp73β, was able to reverse the changes in basal expression brought about by RAC1b knockdown, i.e., it upregulated ECAD and downregulated SLUG mRNA levels (Figure 5D). Moreover, combined silencing of TAp73 and RAC1b (corresponding to panel C) failed to provide an extra decrease in ECAD expression or an extra increase in SLUG expression over that achieved with the RAC1b knockdown alone (Figure 5E).

Together, this suggests that (i) TAp73α and RAC1b act as part of the same signaling pathway and that (ii) TAp73α operates downstream of RAC1b to mediate its negative effects on cell motility and corresponding effects on EMT marker expression, thereby confirming the expression data from Figure 4, and (iii) RAC1b may utilize an additional mechanism(s) to inhibit migration that is not shared by TAp73.

### 3.5. Inhibition of BGN Synthesis or Its Depletion from the Medium Reproduced the Promigratory Effect of RAC1b or TAp73 Silencing and Is Associated with Increased ERK Activation

Above, we have shown that both RAC1b and TAp73 collaborate in inhibiting migration and that RAC1b—like TAp73—induces SMAD4 [15] and BGN [16]. This is consistent with earlier data from Thakur and colleagues from a GEMM showing that the loss of *TP73* and, as a consequence, defective production and secretion of Bgn caused EMT and enhanced migration in the murine PDAC cells. To reveal whether in human cells the antimigratory effects of RAC1b and TAp73 converge on BGN, we silenced BGN expression by RNAi and subjected cells to cell migration assays in the absence or presence of recombinant human TGF-β1. As predicted, we observed increased basal migratory activity in PANC-1 cells as a result of knocking down *BGN* (Figure 6A, green curve/tracing C vs. red curve/tracing A). Moreover, the stimulatory effect of TGF-β1 was enhanced in BGN-knockdown cells to a much greater extent than in control cells (Figure 6A, magenta curve/tracing D vs. blue curve/tracing B). Conversely, ectopic expression of BGN was able to suppress not only basal but also TGF-β1-induced cell migration (Figure 6B, magenta curve/tracing D vs. blue curve/tracing B).

Next, we used an anti-BGN antibody raised against a peptide within the mature form of human BGN (LF-51) to deplete secreted BGN from the conditioned medium of PANC-1 cells. In the presence of this antibody, but not an isotype control antibody, PANC-1 cells migrated more vigorously (Figure 6B, red curve/tracing B vs. green curve/tracing A). Together, the data so far confirm the inhibitory role of secreted BGN on basal and TGF-β-driven cell motility.

Previously, we had shown that both RAC1b and TAp73 are potent inhibitors of basal and TGF-β1-induced ERK activation [15,17], a major pathway driving EMT and cell motility in PDAC [11]. To find out whether this effect is mediated by BGN, we analyzed the same cells from Figure 6A by immunoblotting for ERK1/2 phosphorylation. Intriguingly and in agreement with our assumption, the activated forms of ERK1 and ERK2 were much more abundant in BGN-knockdown cells when compared with control cells (Figure 6D and Appendix A). Collectively, these data establish a clear connection between BGN and the observed phenotypes and strongly suggest that the RAC1b-TAp73-SMAD4 pathway exerts its anti-migratory effect in human PDAC cells through (secreted) BGN-mediated inhibition of ERK activation.

### 3.6. Identification of Downstream Targets and Signaling Pathways That Are Affected by RAC1b, TAp73 and BGN in the Same Direction

In previous studies, we demonstrated that both RAC1b [17], human TAp73 [15] and murine TAp73 [14] control various EMT markers, such as ECAD, VIM and SNAIL. Given the importance of the EMT process for tumor progression, it was of interest if BGN, too, as a common downstream effector impacts these targets in the same way as RAC1b and TAp73. We have thus analyzed PANC-1 cells with RNAi-mediated knockdown, or ectopic overexpression, of BGN, and RNAi-mediated knockdown of p73 as a control, for expression of ECAD, VIM and SNAIL by immunoblotting. We observed a decreased abundance of ECAD but an increased one for VIM and SNAIL (Figure 7A and Appendix A). Conversely, after ectopic expression of BGN, we noted upregulation of ECAD and downregulation of VIM and SNAIL (Figure 7B and Appendix A).

Recently, Cave and colleagues demonstrated that the overexpression of LAMC2 induced by TGF-β1 improves the tumorigenic potential of the PDAC cells both in vitro and in vivo [20]. We, therefore, evaluated the expression of LAMC2 in PANC-1 cells transfected with either empty vectors, or expression vectors encoding either TAp73α or TAp73β (Figure 7C). We found that transfection of both the α and the β isoform of TAp73 provided an increase in basal expression of LAMC2 as did a plasmid encoding TGF-β1 (Figure 7C, white bars). Upon stimulation of the cells with exogenous TGF-β1, LAMC2 was induced in empty vector control cells and in TAp73α transfectants but not in the TAp73β transfected counterparts (Figure 7C, black-filled vs. white bars). These data suggest that LAMC2 is induced by TGF-β1 via TA-p73-SMAD4 rather than Smad-independent signaling.

With respect to signal transduction, we have shown that RAC1b [17], TAp73 [15] and BGN (this study, Figure 6D) negatively regulate the activation of the MEK-ERK pathway. In light of the Thakur study highlighting stromal BGN as a TGF-β inhibitor and tumor suppressor in a murine model of human PDAC, we were especially interested in revealing if all three proteins do affect TGF-β signaling. Previous data from our group revealed that RAC1b negatively controls TGF-β signaling via the ALK2-SMAD1/5 arm [21] with inhibition of RAC1b in PANC-1 cells via RNAi-mediated knockdown or genomic knockout causing elevated levels of C-terminally phosphorylated SMAD1/5 (pSMAD1/5) after TGF-β1 treatment. In order to reveal whether TAp73 or BGN can mimic this effect, we transfected PANC-1 cells with siRNA to p73 or BGN and after a 1 h stimulation with TGF-β1 measured the abundance of pSMAD1/5 by phospho-immunoblotting. Of note, the knockdown of either TAp73 or BGN resulted in elevated levels of pSMAD1/5 but not pSMAD3 (Figure 7D and Appendix A). Together, this shows that BGN phenocopied the effects of RAC1b or TAp73 on the above downstream targets and, in addition, that all three proteins selectively interfere with the ALK2-SMAD1/5 arm of TGF-β signaling. In essence, this confirms our contention that they are part of the same tumor suppressor pathway.

## 4. Discussion

Although some progress has recently been achieved in the treatment of PDAC, the prognosis for patients suffering from this cancer type is still dismal. A better understanding of the molecular events driving tumor development and progression is thus of utmost importance. The p53 homolog, TAp73, has been reported to be involved in cancer development through regulating cell proliferation and apoptosis. Several studies have confirmed the crosstalk of the p53 and the TGF-β networks [22], two major regulators of cancer-associated pathways. In PDAC, their interplay seems to be associated with SMAD proteins, in particular SMAD4 [23]. The deletion or mutation of the SMAD4-encoding *DPC4* correlates with shorter survival and widespread metastasis. However, a similar interplay between TAp73 and TGF-β signaling has only later been revealed in a pioneering study by Thakur and coworkers. Using GEMMs, these authors have shown that TAp73-deficient PDAC exhibited enhanced desmoplasia and characteristics of EMT, including increased migratory/invasive capacity and drug resistance, suggesting enhanced activity of TGF-β [14]. The absence of TAp73 also led to a decrease in SMAD protein levels resulting in a failure to activate the SMAD-dependent pathway and to induce expression of the TGF-β/Smad target, Bgn. As a result of the absence of TGF-β binding and neutralization by Bgn, elevated levels of free TGF-β accumulate in the tumor cells. Of note, high serum TGF-β1 has been proposed to be linked to an increased risk of pancreatic cancer [24]. Moreover, in a GEMM of metastatic breast cancer TGF-β induced by anticancer treatment has been identified as a pro-metastatic signal in tumor cells [25]. In the absence of Smad4 and Bgn, TGF-β signaling switches to Smad-independent pathway activation. The derepression of non-Smad, i.e., ERK and PI3K/AKT signaling in TAp73 deficient cells [14], favors the expression of EMT-associated transcription factors and thus promotes EMT and invasion. The data presented in the Thakur study with the TAp73-deficient GEMM suggest that TAp73 efficiently prevents a switch in TGF-β function from tumor-suppressive to tumor-promoting and that this switch also involves secreted factors acting in an autocrine/paracrine fashion. However, whether this TAp73-driven pathway also operates in human PDAC has remained unresolved so far. In recent work, we were able to show in the human PDAC cell lines PANC-1 and HPAFII that TAp73α [15] antagonized EMT by upregulating basal and TGF-β1-induced expression of epithelial markers, like ECAD, and downregulating that of mesenchymal markers, like SNAIL, and non-Smad, i.e., ERK1/2 signaling [15]. Moreover, TAp73 exhibited a strong antimigratory effect on these cells consistent with its anti-EMT function. This induction of epithelial or anti-mesenchymal genes with simultaneous suppression of mesenchymal genes and pathways was reminiscent of what we observed earlier for RAC1b [17]. As shown here for TAp73, RAC1b, too, induces the expression of the TGF-β inhibitor, BGN [16]. In concordance with the crucial of Bgn in mediating the tumor-suppressive effect of TAp73 in *Tp73*-deficient mice, we provide evidence in the present study that in human PDAC, RAC1b and TAp73α collaborate in promoting the expression of BGN and, as a consequence, inhibit basal and TGF-β-driven ERK activation and cell migration. Specifically, we have identified BGN as a paracrine effector of RAC1b and TAp73 in human pancreatic cancer cells by showing that the basal and TGF-β1-driven ERK activating and promigratory effects of RAC1b/*RAC1* exon 3b silencing [17,18], TAp73 silencing [15] or SMAD4 silencing [15] are duplicated by gene silencing of *BGN* or by antibody-mediated neutralization of its biological activity in culture supernatants.

A major goal of this study was to evaluate if TAp73 and RAC1b are members of the same tumor-suppressive pathway. Of note, we have previously established a RAC1b-SMAD3-BGN axis in PANC-1 cells to be critical in maintaining the epithelial phenotype, already suggesting the possibility that RAC1b may be able to induce other Smad proteins besides SMAD3 as shown previously for TAp73 in mice [14]. In the present study, we found that in human PDAC cells RAC1b, indeed, positively controls SMAD4 expression at both the mRNA and protein levels. Given the preferred association of RAC1b expression in PDAC with cells of an epithelial subtype [16], this suggested the possibility that SMAD4 (and SMAD3) contribute to both maintenance of the epithelial phenotype and tumor suppression and that TAp73 and RAC1b act upstream of these SMADs as part of the same pathway to control their expression (Figure 8). Having shown that RAC1b and TAp73 collaborate in pro-differentiation, anti-EMT and anti-migration effects in PDAC-derived cells by promoting SMAD3/4 and BGN expression and inhibiting ERK activation, we wished to gain insight into the downstream targets and signaling pathways that TAp73, RAC1b or BGN affect. Following RNAi-mediated knockdown or ectopic overexpression of BGN in PANC-1 cells, we found that BGN phenocopied the effects of RAC1b and TAp73 on the EMT markers ECAD, VIM and SNAIL (Figure 7A,B), and on the ALK2-SMAD1/5 arm of TGF-β signaling (Figure 7C). This means that RAC1b, TAp73 and BGN control EMT marker expression and TGF-β1 signaling via SMAD1/5 in the same way, which confirms our contention that all three proteins are part of the same signaling pathway. Hence, we provide proof for the existence of a RAC1b-TAp73α-SMAD4-BGN axis operating in both murine and human cells to provide tumor suppression by maintaining epithelial differentiation. Moreover, this is the first demonstration of RAC1b collaborating with an established tumor suppressor pathway.

We also sought to know whether RAC1b is located upstream or downstream of TAp73. In a series of expression experiments with reciprocal inhibition and overexpression or mutual rescue experiments with migratory activities as readout, we came to the conclusion that RAC1b acts upstream of TAp73. If RAC1b is to activate TAp73 then it should be located in the nucleus. In fact, nuclear localization of RAC1b has been reported and compared to the parental isoform, RAC1. Interestingly, RAC1b more strongly accumulates in the nucleus as a result of less prenylation, which in turn is due to a more stable association with SmgGDS-607 [26]. This spatial proximity to TAp73 may explain why RAC1b can induce SMAD4 via TAp73 binding to the *DPC4* promoter. As speculated earlier for murine cells [14], TAp73 deficiency in PANC-1 cells may cause a decrease in transactivation of the *DPC4* promoter harboring a p53 response element [14].

Despite being a mesenchymal matrix protein, BGN is nevertheless subject to positive regulation by both RAC1b [16], TAp73 ([14], this study) and SMAD4 (this study). This is noteworthy as it clearly shows that the type of regulation depends on the protein’s function with respect to EMT (here a TGF-β inhibitor) rather than its general structure. Thakur and colleagues have shown that Bgn, unlike other mesenchymal proteins, is a potent inhibitor of the EMT process in murine cells, likely by its ability to bind TGF-β and neutralize its biological activity [14]. Our data here in human cells are in good agreement with those from the mouse as we have shown that inhibition of BGN expression or its biological activity derepresses the cells’ migratory/invasive activities and that this is associated with activation of non-SMAD/ERK signaling (Figure 8). The strong promoting effect of TAp73 on the induction of *BGN* by TGF-β1 via intermittent SMAD4 expression may thus serve to ensure sufficient BGN production for the neutralization of this growth factor in a TGF-β-rich microenvironment.

Of note, although RAC1b inhibition caused a decrease in SMAD3 and SMAD4 abundance, the overall TGF-β signaling activity was higher, due to the loss of RAC1b-mediated suppression of the synthesis of ALK5 (the major TGF-β type I receptor) (Figure 8). Notably, this function of RAC1b was not shared by TAp73 (H.U., unpublished observation), which we interpret as additional evidence for RAC1b being located upstream of TAp73, because in case of the reverse orientation, TAp73 would be expected to also impair ALK5 expression and signaling.

Altogether, these data clearly suggest that the absence of RAC1b or TAp73 impairs TGF-β signaling toward the tumor-suppressing SMAD4-dependent pathway. Hence, the collaboration between RAC1b and TAp73 in suppressing EMT and cell motility might extend to other tumor-suppressive modes of TGF-β; for instance, a SMAD4-dependent lethal form of EMT [27]. Also, in vivo data in mouse models will reveal if TAp73 or RAC1b deficiency will reduce the number of liver metastases developing from the cells after their injection into the pancreas, and—mechanistically—if the resulting mesenchymal conversion accounts for the pro-metastatic effect.

## 5. Conclusions

The results of this study may have implications for therapeutically targeting TGF-β in patients. A series of clinical trials with agents that inhibit either the TGF-β ligand or the receptors are currently in progress [28]. However, PDAC treatment with these inhibitors is challenging due to the lack of predictive biomarkers that aid in identifying those patients that are likely to respond. The availability of those biomarkers may facilitate the selection of patients and the optimal time for treatment with regard to TGF-β activity. There is also evidence from other cancer types that RAC1b [29,30], TAp73 [31,32] and BGN [33,34,35] all promote chemoresistance and that they may even operate through the same downstream signaling, i.e., NF-κB [30,33]. Future studies should reveal if the levels or activation states of TAp73 and/or RAC1b in patients could aid in assessing whether the oncogenic- or tumor-suppressive actions of TGF-β predominate at a given time.

## Figures and Tables

**Figure 1 biomedicines-12-00199-f001:**
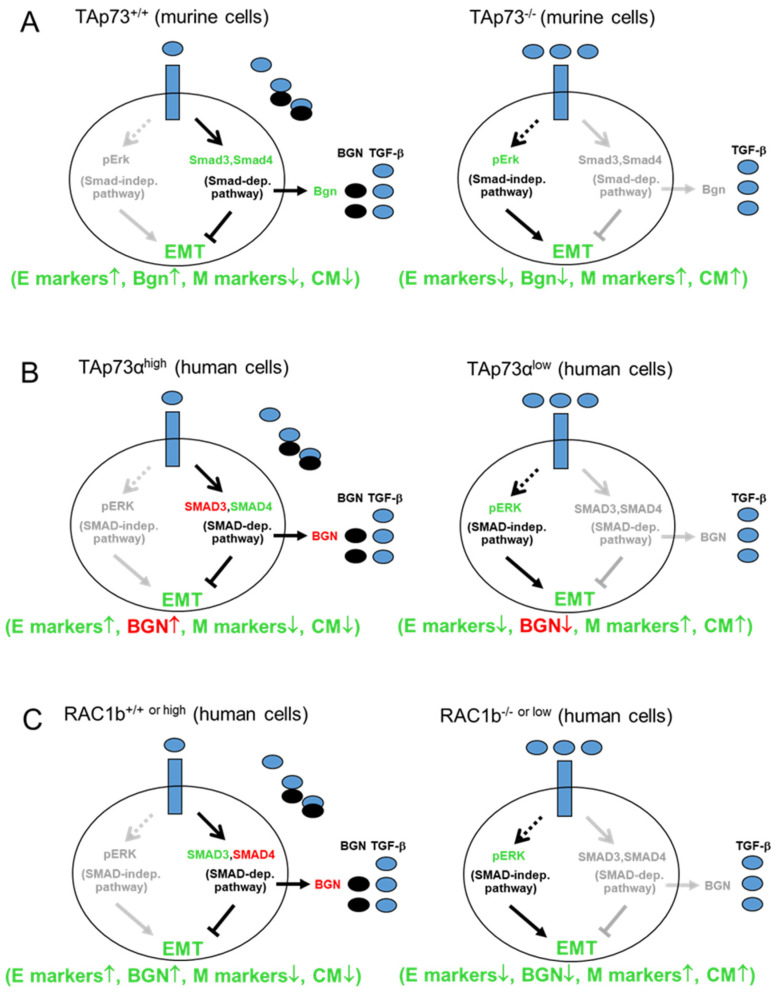
Scheme illustrating the current knowledge on tumor-suppressive signaling of (**A**) TAp73 in murine PDAC cells and (**B**) TAp73α in human PDAC cells and (**C**) RAC1b in human PDAC cells. The regulatory interactions shown in (**A**) were established in mouse models of pancreatic cancer [14]. In the presence of TAp73 (left-hand cartoon) and following activation of TGF-β receptors (blue rectangles) by TGF-β (blue ovals), the Smad-dependent pathway is activated. This promotes tumor-suppressive functions including Bgn secretion (black ovals), which traps TGF-β within the ECM of the pericellular space, to largely prevent receptor binding, activation of non-Smad, i.e., Erk1/2 signaling and the induction of EMT. In the absence of TAp73 (right-hand cartoon), the expression/activity of Smad proteins is reduced and as a consequence, Bgn is no longer secreted, limiting its TGF-β trapping within the ECM. This creates a positive loop that reinforces the oncogenic impact of Smad-independent pathways and stimulation of EMT. (**B**) Some of the proteins shown in (**A**) have been analyzed and confirmed recently in human PDAC cells with knocked down or ectopically expressed TAp73α ([15], marked in green lettering). Yet the roles of others (denoted in red lettering) such as SMAD3 (not subject of the present study) and expression and secretion of BGN remain to be demonstrated. (**C**) In contrast, most of the interactions shown in (**A**) have been established in human PDAC cells with either genomic deletion of the RAC1b-encoding exon 3b of *RAC1* (−/−) or knockdown (low) of RAC1b ([16,17,18], denoted in green lettering), except for SMAD4 and the secreted form of BGN (marked in red lettering). Based on the congruence/similarity of the regulatory interactions between (**A**), (**B**) and (**C**) in both the wildtype/physiological and mutated/inhibited states, we postulate that TAp73 and RAC1b are components of the same tumor suppressor pathway. The arrows indicate induction of expression or activation, while the lines indicate suppression. Black-lettered protein names or black arrows/lines indicate the active state, while the grey-shaded counterparts mark the inactive state. For details see text. CM, cell migration; E, epithelial; M, mesenchymal.

**Figure 2 biomedicines-12-00199-f002:**
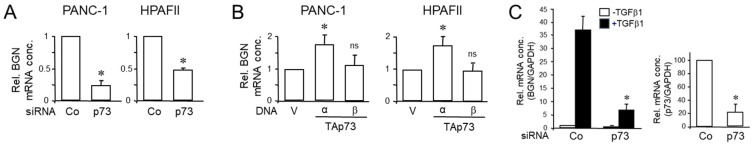
TAp73 is crucial for basal and TGF-β-dependent expression of BGN. (**A**) PANC-1 or HPAFII cells were transfected with 50 nM each of p73 siRNA or an irrelevant control (Co) siRNA on two consecutive days and 48 h after the second round of transfection processed for qPCR analysis of BGN expression, and GAPDH to account for small differences in cDNA input. Data represent the normalized mean ± SD of three assays. (**B**) PANC-1 or HPAFII cells were transfected with either empty vector (V), or expression vectors encoding either TAp73α or TAp73β and subjected to BGN PCR, including GAPDH as an internal control. Data represent the normalized mean ± SD of three assays. (**C**) Knockdown of TAp73 interfered with TGF-β1-induced regulation of *BGN*. PANC-1 cells were exposed for 24 h to rec. human TGF-β1 and subsequently assayed by qPCR for *BGN*, and *TP73* to verify successful knockdown. Data are the mean ± SD of three experiments. The asterisks (*) denote a significant difference compared to the Co siRNA (**A**,**B**, right-hand graph in **C**) or between the two TGF-β1 treated samples in the left-hand graph of the panel (**C**) (*p* < 0.05, Wilcoxon rank-sum test). ns, non-significant. Successful knockdown of p73 and ectopic expression of TAp73α and TAp73β had been verified previously by qPCR analyses and immunoblotting [15].

**Figure 3 biomedicines-12-00199-f003:**
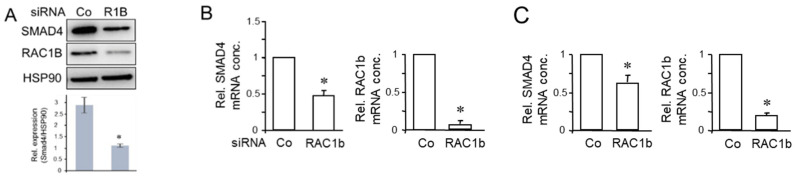
RAC1b promotes the expression of SMAD4. (**A**) PANC-1 cells were transfected with either RAC1b (R1B) siRNA or an irrelevant control (Co) siRNA (50 nM each) and 48 h later processed for sequential immunoblotting of SMAD4, RAC1b and HSP90 as a loading control. The graph below the blot depicts results from densitometry-based quantification of the SMAD4 protein (mean ± SD, n = 3). (**B**) As in (**A**), except that cells were processed for qPCR analysis of SMAD4 (left-hand graph), and RAC1b as a control for successful knockdown (right-hand graph). (**C**) As in (**B**), except that L3.6pl cells were used. Data represent the mean from three assays (mean ± SD) after normalization with GAPDH. Asterisks (*) indicate significant differences relative to the Co siRNA (*p* < 0.05, Wilcoxon rank-sum test).

**Figure 4 biomedicines-12-00199-f004:**
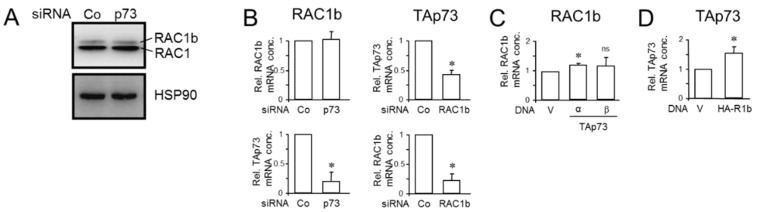
Knockdown of RAC1b decreases TAp73 expression but not vice versa. (**A**) PANC-1 cells were subjected to two rounds of transfection (on two consecutive days) with 50 nM each of TAp73 siRNA or control siRNA and processed for immunoblotting of RAC1b (upper band) and RAC1 (lower band). The blot was also probed with an antibody to HSP90 to test for equal protein loading. Successful knockdown of p73 was verified by qPCR analysis as shown in (**B**) and earlier by immunoblotting [15]. (**B**) PANC-1 cells were transiently transfected with 50 nM of siRNA to either RAC1b, TAp73, or a scrambled Co siRNA, and subsequently subjected to qPCR analyses of RAC1b (left two graphs) or TAp73 (right two graphs). (**C**,**D**) RAC1b increases TAp73 mRNA abundance but not vice versa. (**C**) PANC-1 cells transiently transfected with either empty vector (V) or HA-tagged versions of either TAp73α or TAp73β were analyzed by qPCR for expression of RAC1b. (**D**) As in (**C**), except that PANC-1 cells stably expressing either empty pCGN vector (V) or HA-RAC1b in pCGN (HA-R1b) were analyzed by qPCR for expression of TAp73. In (**C**,**D**), data represent the mean ± SD (n = 3). Asterisks (*) indicate significance relative to the Co siRNA (**B**) or the empty vector (**C**,**D**) (*p* < 0.05, Wilcoxon rank-sum test); ns, non-significant.

**Figure 5 biomedicines-12-00199-f005:**
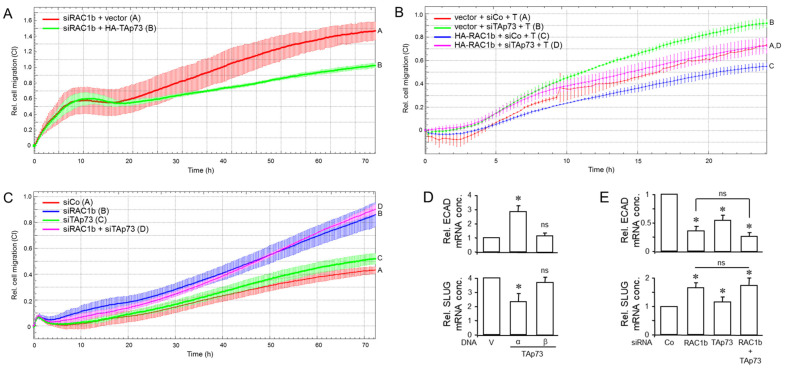
Mutual rescue experiments with various combinations of RAC1b and TAp73 knockdown and ectopic overexpression approaches. (**A**) PANC-1 cells were transfected with either RAC1b siRNA (siRAC1b) in combination with TAp73α (HA-TAp73) or empty pcDNA3.1 vector (vector). (**B**) As in (**A**), except that PANC-1 cells received empty pCGN vector or HA-RAC1b-pCGN in combination with siRNA to p73 (sip73) or scrambled control siRNA (siCo). (**C**) PANC-1 cells were transfected with 50 nM of siCo, or 25 nM each of siRAC1b and siCo, sip73 and siCo, or a combination of 25 nM each of siRAC1b and sip73. Representative assays out of three assays performed in total for each condition are shown. Data are the mean ± SD from 3–4 parallel wells. Successful ectopic expression of TAp73α and siRNA-mediated knockdown of TAp73 protein was shown previously [15], while RNAi-mediated inhibition of RAC1b protein is presented in Figure 3A. (**D**) The same cells from (**A**), and, additionally, PANC-1 cells transfected with TAp73β, were monitored by qPCR analysis for expression of ECAD (upper graph) and SLUG (lower graph). Data are displayed relative to cells transfected with RAC1b siRNA + vector set arbitrarily at 1.0. (**E**) The same cells from (**C**) were screened by qPCR analysis for expression of ECAD (upper graph) and SLUG (lower graph). Data in (**D**,**E**) represent the mean ± SD of three assays (n = 3). The asterisks (*) indicate significance relative to vector or siRNA controls (*p* < 0.05, Wilcoxon rank-sum test); ns, non-significant.

**Figure 6 biomedicines-12-00199-f006:**
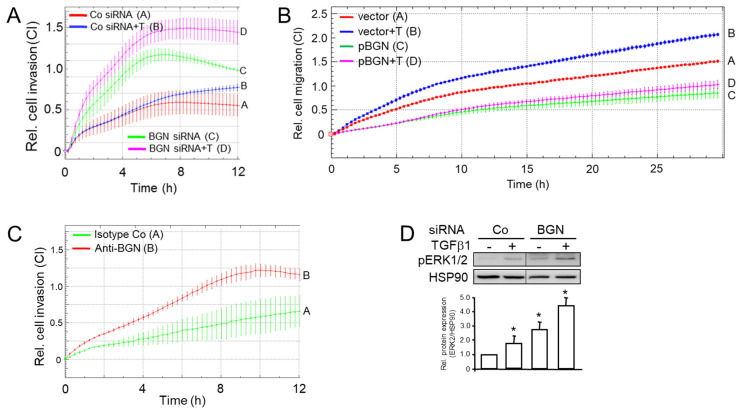
Real-time cell migration assay of PANC-1 cells after RNAi-mediated knockdown of BGN. (**A**) PANC-1 cells were transfected with 50 nM of BGN siRNA or an irrelevant siRNA (Co siRNA) and subjected to real-time cell migration assay on an xCELLigence platform in the absence or presence of TGF-β1 (T, 5 ng/mL). Data are the mean ± SD of quadruplicate wells. The assay is representative of three independent assays. Successful knockdown of BGN in PANC-1 cells has been validated by qPCR [16]. (**B**) PANC-1 cells were transfected with an empty vector (pcDNA3.1) or the same vector encoding human BGN (pBGN) as outlined in the Methods section. Forty-eight h after the start of transfection, cells were detached, counted and equal numbers of cells were analyzed for migratory activity as described under (**A**). Successful overexpression of BGN has been verified by qPCR (Appendix A). (**C**) Migration assay of PANC-1 cells treated with an anti-BGN antibody (LF-51), or isotype control antibody, during the course of the assay. Data are the mean ± SD of quadruplicate wells from a representative experiment. (**D**) Immunoblot analysis of PANC-1 cells transfected as described in (**A**) and treated, or not, for 1 h with TGF-β1 (5 ng/mL). The blot was probed with antibodies to phospho-ERK1/2 (pERK1/2), and to HSP90 as a loading control. The graph below the blot depicts results from densitometric readings of signal intensities from underexposed autoradiograms (mean ± SD, n = 3). The asterisks (*) in (**D**) indicate significance relative to the non-TGF-β-treated Co (*p* < 0.05, Wilcoxon rank-sum test). The vertical lines between bands denote the removal of irrelevant lanes.

**Figure 7 biomedicines-12-00199-f007:**
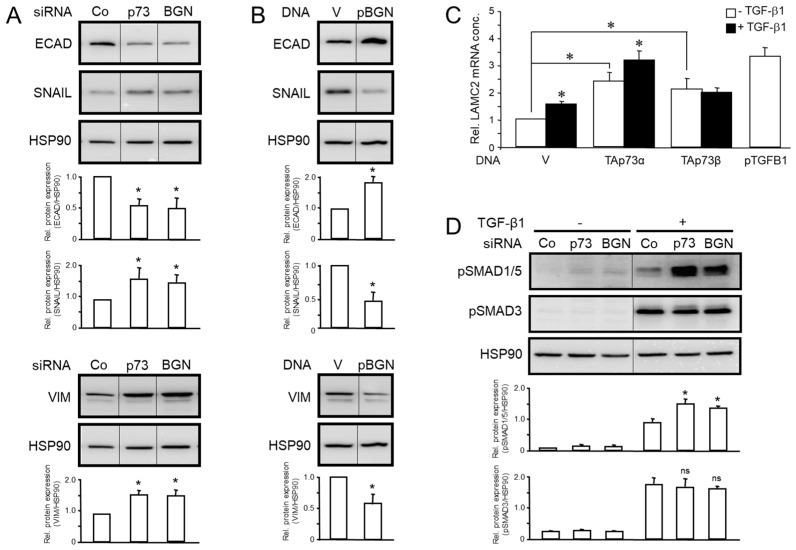
Identification of downstream targets and signaling pathways that are shared by BGN and TAp73. (**A**) PANC-1 cells were transfected with 50 nM of siRNA specific for p73 or BGN, or with an irrelevant control (Co) siRNA. Forty-eight h later, cells were processed for sequential immunoblotting of ECAD, SNAIL and VIM, and HSP90 as a loading control. The graphs underneath the blots show the results from relative protein quantification based on densitometric readings of band intensities (mean ± SD, n = 3). (**B**) As in (**A**), except that cells were transfected with either a BGN expression vector (pBGN) or empty vector (V) as control. The graphs depict results from densitometry-based protein quantification (mean ± SD of three independent experiments). (**C**) PANC-1 cells were transfected with empty vector (V), or expression vectors for TAp73α, TAp73β or TGF-β1 (pTGFB1). Twenty-four h later, cells were processed for qPCR analysis of LAMC2 (mean ± SD, n = 3). (**D**) PANC-1 cells were transfected with the same siRNAs as in (**A**) and 48 h after transfection were either left untreated or were treated with 5 ng/mL TGF-β1 for 1 h. Following lysis, cells were subjected to sequential phospho-immunoblotting of pSMAD1/5 and pSMAD3, and HSP90 to control for equal loading. The graphical data shown represent the mean and SD from three experiments after normalization with HSP90. The asterisks (*) in (**A**,**D**) indicate significant differences relative to the Co siRNA, and those in (**B**) significant differences relative to the V control. In (**C**), the asterisks over black-filled bars denote a significant difference relative to the respective non-TGF-β1 treated vector control (*p* < 0.05, Wilcoxon rank-sum test); ns, non-significant.

**Figure 8 biomedicines-12-00199-f008:**
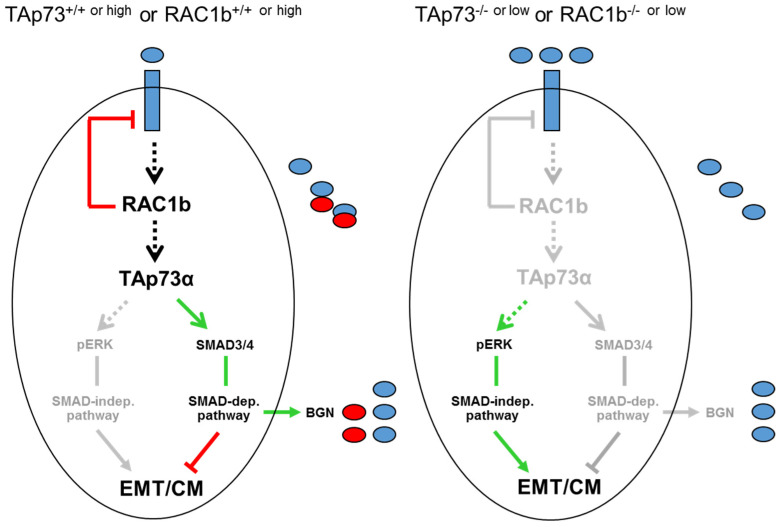
Combined version of the cartoons shown in Figure 1 to illustrate the role of TAp73 and RAC1b in negative control of EMT, cell migration (CM) and TGF-β activity in human PDAC cells. Upon exposure of TAp73^+/+^ or RAC1b^+/+ or high^ epithelial tumor cells to EMT inducers such as TGF-β1 (blue ovals, left-hand scheme), RAC1b increases the level of or activate TAp73α to induce SMAD3/4 expression and signaling, BGN secretion (red ovals) and, ultimately, inhibition of TGF-β and TGF-β-induced activation of the ERK pathway. A decrease in activation of the SMAD pathway, i.e., by limiting the synthesis of SMAD4 following RAC1b or TAp73 inhibition (right-hand scheme, TAp73^−/− or low^ or RAC1b^−/− or low^) induced a switch in TGF-β signaling to SMAD4-independent pathways, e.g., activation of ERK1/2. In the course of this study, we have also carved out that RAC1b is located upstream of TAp73 in this pathway and that the α rather than the β isoform of TAp73 is controlling BGN. Unlike TAp73α, RAC1b is a potent suppressor of ALK5 expression (indicated by the red line between RAC1b and the blue-filled rectangle in the left-hand scheme) and this function overrides the increase in TGF-β responsiveness otherwise awarded by TAp73α. The green arrows denote activation and the red lines suppression. Arrows and lines in grey mark the inactive state. For details see text.

## Data Availability

All data reported are contained in Section 3 and the Appendix A.

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
