# Peer review of "RAC1b Collaborates with TAp73α-SMAD4 Signaling to Induce Biglycan Expression and Inhibit Basal and TGF-β-Driven Cell Motility in Human Pancreatic Cancer"

_biomedicines, 2024, doi:10.3390/biomedicines12010199_

Round 1

Reviewer 1 Report

Comments and Suggestions for Authors

Pancreatic ductal adenocarcinoma (PDAC) belongs to the very aggressive tumors with limited therapeutic options. The TGF-beta pathway is essential in PDAC development, but its regulation should be better described.

The research by Ungefroren et al. showed that RAC1b and transcriptionally active p73 alfa (TAp73alfa) are  SMAD3/4 signaling pathway activators. SMAD3/4 signaling is engaged in the secretion of biglycan and, subsequently, in inhibiting TGF-beta and related epithelial-mesenchymal transition, migration, and invasion.

Several issues in the paper should be improved:

1. The Abstract is significantly too long; it should be up to 200 words.

2. The name of genes should always be written in italics, e.g., line 185.

3. The sharpness of Figure 2 is partly incorrect.

4. The value from line 248  and Abstract (48.3 ± 6.7-fold to 8.8 ± 3.2-fold) looks different from the data in Figure 2C.

Author Response

Dear Editor, dear Alexandra,

Please find attached the revised version of our latest manuscript, biomedicines-2758355. We thank the reviewers for their constructive critizisms and have done our best to satisfy their comments. We are convinced that this has considerably increased the quality of our paper. All revisions to the manuscript have been highlighted in the “track changes” mode. Moreover, the highlighted sentences have been partly modified with the “track changes” mode to remove duplications in the course of shortening the Abstract (a request of Reviewer 1).

Pancreatic ductal adenocarcinoma (PDAC) belongs to the very aggressive tumors with limited therapeutic options. The TGF-beta pathway is essential in PDAC development, but its regulation should be better described.

The research by Ungefroren et al. showed that RAC1b and transcriptionally active p73 alfa (TAp73alfa) are SMAD3/4 signaling pathway activators. SMAD3/4 signaling is engaged in the secretion of biglycan and, subsequently, in inhibiting TGF-beta and related epithelial-mesenchymal transition, migration, and invasion.

Several issues in the paper should be improved:

  1. The Abstract is significantly too long; it should be up to 200 words.

Response: As requested, the abstract has been condensed to 254 words. We feel that further shortening will spare important information for the readership.

  1. The name of genes should always be written in italics, e.g., line 185.

Response: Genes names have been italicized.

  1. The sharpness of Figure 2 is partly incorrect.

Response: As requested, the sharpness of all panels in this Figure has been restored.

  1. The value from line 248 and Abstract (48.3 ± 6.7-fold to 8.8 ± 3.2-fold) looks different from the data in Figure 2C.

Response: The values given in the Abstract have been deleted in the course of shortening the abstract.

Reviewer 2 Report

Comments and Suggestions for Authors

Manuscript entitled "RAC1b Collaborates with TAp73α-SMAD4 Signaling to Induce Biglycan Expression and Inhibit Basal and TGF-β-Driven Cell Motility in Human Pancreatic Cancer"

Several issues:

1. There is no "direct mechanism" disclosed. There are some correlations and phenotypes, but no direct link of the targets. The authors should performed Co-IP, ChIP, ... etc. to confirm a direct link.

2. There is no clinical relevance disclosed. The  authors should validate the associations of these biomarkers and their clinical significance, and survival impacts in a well collected in-house cohort.

Comments on the Quality of English Language

acceptable.

Author Response

Dear Editor, dear Alexandra,

Please find attached the revised version of our latest manuscript, biomedicines-2758355. We thank the reviewers for their constructive critizisms and have done our best to satisfy their comments. We are convinced that this has considerably increased the quality of our paper. All revisions to the manuscript have been highlighted in the “track changes” mode. Moreover, the highlighted sentences have been partly modified with the “track changes” mode to remove duplications in the course of shortening the Abstract (a request of Reviewer 1).

Manuscript entitled "RAC1b Collaborates with TAp73α-SMAD4 Signaling to Induce Biglycan Expression and Inhibit Basal and TGF-β-Driven Cell Motility in Human Pancreatic Cancer"

Several issues:

  1. There is no "direct mechanism" disclosed. There are some correlations and phenotypes, but no direct link of the targets. The authors should perform Co-IP, ChIP, ... etc. to confirm a direct link.

Response: We partly agree with the reviewer on this point. We did show, however, that TAp73 is directly regulated by RAC1b via a (post-)transcriptional mechanism. We believe that clarifying its precise type, e.g., enhanced promoter/transcriptional activity or (post-trancriptional) changes in mRNA stability, would require a large set of additional experiments that would make the manuscript too long and maybe somewhat beyond the scope of this study. Rather, we plan to study the mechanistic aspects in more detail and publish these data in a separate manuscript.

  1. There is no clinical relevance disclosed. The authors should validate the associations of these biomarkers and their clinical significance, and survival impacts in a well collected in-house cohort.

Response: Again, this is a very good suggestion for future studies. Indeed, we are envisaging such a study. Unfortunately, we do not currently have access to so many patient data to perform such validations.

Reviewer 3 Report

Comments and Suggestions for Authors

·    The authors demonstrated that in human PDAC cells TAp73a is necessary for both basal expression of BGN and its full-blown response to TGF-B1. Recently, Delle Cave and colleagues demonstrated that the overexpression of LAMC2 induced by TGF-B1 improves the tumorigenic potential of the PDAC cells both in vitro and in vivo (doi: 10.1186/s13046-022-02516-w). The authors must evaluate the expression of LAMC2 in PANC-1 or HPAFII cells transfected with either empty vector (V), or expression vectors encoding either TAp73a or TAp73.

·       The authors demonstrated that the increase in Cell Migration Upon RAC1b Knockdown is Partially Rescued by Ectopic Expression of TAp73. Since migration is also related to in epithelial-mesenchymal transition (EMT) the authors should evaluate the levels of EMT markers (such as e-cadherin, vimentin, snail1, twist1) in all the cellular systems described.

Comments on the Quality of English Language

good

Author Response

Dear Editor, dear Alexandra,

Please find attached the revised version of our latest manuscript, biomedicines-2758355. We thank the reviewers for their constructive critizisms and have done our best to satisfy their comments. We are convinced that this has considerably increased the quality of our paper. All revisions to the manuscript have been highlighted in the “track changes” mode. Moreover, the highlighted sentences have been partly modified with the “track changes” mode to remove duplications in the course of shortening the Abstract (a request of Reviewer 1).

The authors demonstrated that in human PDAC cells TAp73a is necessary for both basal expression of BGN and its full-blown response to TGF-B1. Recently, Delle Cave and colleagues demonstrated that the overexpression of LAMC2 induced by TGF-B1 improves the tumorigenic potential of the PDAC cells both in vitro and in vivo (doi: 10.1186/s13046-022-02516-w). The authors must evaluate the expression of LAMC2 in PANC-1 or HPAFII cells transfected with either empty vector (V), or expression vectors encoding either TAp73a or TAp73.

Response: We thank the reviewer for this nice suggestion. As requested, we have performed qPCR analysis of LAMC2 in PANC-1 cells and found that i) both ectopic TAp73α and TAp73β increased LAMC2 mRNA levels and ii) that stimulation of cells with exogenous TGF-β1 was able to induce LAMC2 expression in vector controls and in TAp73α but not TAp73β transfected cells. These data are now shown in Figure 7, panel C. The former panel C has become panel D. 

       The authors demonstrated that the increase in Cell Migration Upon RAC1b Knockdown is Partially Rescued by Ectopic Expression of TAp73. Since migration is also related to in epithelial-mesenchymal transition (EMT) the authors should evaluate the levels of EMT markers (such as e-cadherin, vimentin, snail1, twist1) in all the cellular systems described.

Response: This is a good suggestion. As requested, we evaluated the levels of ECAD and SLUG/SNAIL2 in PANC-1 cells co-transfected with siRNA to RAC1b and expression vectors for either TAp73α or TAp73β. It turned out that TAp73α, but not TAp73β, was able to reverse the changes in basal expression brought about by RAC1b knockdown (now shown in Figure 5, new panel D). Moreover, combined silencing of TAp73 and RAC1b failed to provide an extra decrease in ECAD expression or an extra increase in SLUG expression over that achieved with the RAC1b knockdown alone (Figure 5, new panel E).

Round 2

Reviewer 2 Report

Comments and Suggestions for Authors

The revision is acceptable.

Comments on the Quality of English Language

acceptable

Reviewer 3 Report

Comments and Suggestions for Authors

accept

Comments on the Quality of English Language

accept